# Full-Endoscopic Lumbar Interbody Fusion for Treating Lumbar Disc Degeneration Involving Disc Height Loss: Technical Report

**DOI:** 10.3390/medicina56090478

**Published:** 2020-09-17

**Authors:** Tsuyoshi Harakuni, Hiroki Iwai, Yasushi Oshima, Hirokazu Inoue, Tomoaki Kitagawa, Hirohiko Inanami, Hisashi Koga

**Affiliations:** 1Department of Neurosurgery, Urasoe General Hospital, 4-16-1 Iso, Urasoe-City 901-2132, Okinawa, Japan; tthara2000@yahoo.co.jp; 2Department of Neurosurgery, Iwai FESS Clinic, 8-18-4 Minamikoiwa Edogawa-ku, Tokyo 133-0056, Japan; h-iwai@iwai.com; 3Department of Orthopaedics, Iwai Orthopaedic Medical Hospital, 8-17-2 Minamikoiwa Edogawa-ku, Tokyo 133-0056, Japan; yoo-tky@umin.ac.jp (Y.O.); hirokazuinoue0218@gmail.com (H.I.); inanamihiro@gmail.com (H.I.); 4Department of Orthopaedic Surgery, Inanami Spine and Joint Hospital, 3-17-5 Higashishinagawa Shinagawa-ku, Tokyo 140-0002, Japan; 5Department of Orthopaedic Surgery, The University of Tokyo, 7-3-1 Hongo Bunkyo-ku, Tokyo 113-8655, Japan; 6Department of Orthopaedic Surgery, Jichi Medical University, 3311-1 Shimotsuke-shi, Tochigi 329-0431, Japan; 7Department of Orthopaedic Surgery, Teikyo University School of Medicine, 2-11-1 Kaga, Itabashi-ku, Tokyo 173-8606, Japan; kitagawa@med.teikyo-u.ac.jp

**Keywords:** lumbar disc degeneration, full-endoscopic spine surgery, lumbar interbody fusion, minimally invasive, radiculopathy, cauda equina syndrome

## Abstract

*Background and Objectives*: Lumbar disc degeneration (LDD) is the main cause of lower back pain and leads to corresponding disc height loss. Although lumbar interbody fusion (LIF) is commonly used for treating LDD, several different treatment strategies are available. We performed a minimally invasive full-endoscopic LIF (FELIF) using a uniportal full-endoscopic system. *Materials and Methods*: FELIF was performed for 12 patients with LDD with disc-height loss using a 4.1 mm working channel endoscope and a newly developed slider for cage insertion. The mean age of the patients was 68.3 years; the patients presented with single vertebral level involvement. The Brandner’s disc index was used for evaluating the postoperative increase in the disc height. Preoperative and postoperative leg pain was evaluated using the numerical rating scale (NRS) score. *Results*: The mean operation time for FELIF was 109.4 min. The mean duration of hospital stay after FELIF was 7.7 days. There were no operative and postoperative complications, even without drainage during the mean follow-up period of 6.2 months (range, 2–10 months). The Brandner’s disc index improved statistically significant (*p* > 0.01). The mean preoperative and postoperative NRS scores were 6.5 and 1.2, respectively. *Conclusions*: FELIF using a 4.1 mm working channel endoscope can be used for treating LDD with disc height loss. Radiculopathy caused by foraminal stenosis was the most suitable operative indication for FELIF.

## 1. Introduction

Lumbar disc degeneration (LDD) may develop owing to several factors, including genetic and environmental factors [1]. For example, the TAQ 1 polymorphism of the vitamin D receptor (VDR) gene is associated with LDD [2]. In contrast, an experimental study proved that biomechanical stress is distinctly associated with LDD [3]. Regardless of the cause of LDD, LDD subsequently induces lower back pain, radiculopathy, and cauda equina syndrome. The main mechanism underlying radiculopathy and cauda equina symptoms is disc height loss. As severe disc height loss is likely to induce the development of central (cauda equina symptoms) and foraminal (radiculopathy) lumbar spinal canal stenosis (LSCS), increasing the disc height seems to be a logically valid treatment strategy for patients with LDD with disc height loss [4]. Although lumbar interbody fusion (LIF) is the established strategy for increasing the disc height, LIF can be performed using different approaches, such as anterior, posterior, lateral, and transforaminal LIF [5].

Previously, LIF was performed using open, microscopic, or endoscope-assisted surgical approaches. In contrast, uniportal full-endoscopic spine surgery (FESS) was originally introduced for the treatment of lumbar disc herniation and has recently been performed during posterior, lateral, and transforaminal LIF [6,7,8,9,10,11,12,13,14,15,16,17,18,19]. Among these approaches, transforaminal LIF through Kambin’s triangle seems to be the most commonly used approach. According to a systematic review on transforaminal LIF using minimally invasive spine surgery (MIS-TLIF) techniques, FESS through Kambin’s triangle only accounted for 1% of all MIS-TLIF procedures performed [20].

Each investigator has proposed his/her own operative procedures, and there is no consensus regarding the operative indications or procedures presently. Circumstances vary between countries. For example, recombinant biomaterial alternatives to bone grafting are available in the USA [8,15,19] but not in Japan. Biportal FESS has been extensively applied for LIF in Korea [6,11,12], but a 3.9 mm or 4.1 mm working channel endoscope for uniportal FESS has been widely used in Japan. We therefore established a transforaminal LIF procedure using one of these endoscopic systems suitable for the Japanese environment and discussed the advantages and disadvantages of this new procedure.

## 2. Materials and Methods

### 2.1. Patient Selection

Twelve consecutive patients with LDD with disc height loss underwent full-endoscopic LIF (FELIF) using a uniportal FESS system (4.1 mm working channel, RIWOspine GmbH, Knittlingen, Germany) between October 2019 and July 2020. All the patients had lower back pain, radiculopathy, and/or cauda equina syndrome resistant to medical treatment, epidural steroids, and/or nerve block. Neurological examination, preoperative computed tomography (CT), and T2-weighted magnetic resonance imaging (MRI) were performed to identify the location of the LDD with disc height loss and the target area for direct/indirect decompression or stabilization. All the patients had LDD, with disc height loss at only one vertebral level.

The characteristics of the patients and their treatment, including age, sex, symptoms, approach side, operated vertebral level, operative time, duration of postoperative hospital stay, and follow-up period, were obtained from the medical records (Table 1). The extent of decompression was evaluated using the preoperative and postoperative CT and MRI scans. The preoperative and postoperative plain radiographs were used for calculating the Brandner’s disc index (Figure 1) [21]. Preoperative and postoperative pain in the legs was evaluated using the numerical rating scale (NRS) score. The postoperative NRS score was obtained at discharge from the hospital. Statistical analysis was performed using Student’s *t*-test. *p*-values less than 0.05 were considered statistically significant.

### 2.2. Surgical Technique

The patients were carefully logrolled into the prone position. Surgery was performed under general anesthesia, combined with motor-evoked potential monitoring. During surgery, a fluoroscope was placed across the center of the operative table to ensure the appropriate timing of X-ray. The basic operative procedure for the posterolateral approach of uniportal FESS has been described previously [22,23].

First, an ipsilateral percutaneous pedicle screw (PPS; three different types of PPS were used, but RELINE^®^ (NuVasive, San Diego, CA, USA) was used for 10/12 cases) was placed under fluoroscopic guidance. Secondly, an 8 mm skin incision was created 55–90 mm laterally from the midline (posterolateral approach). After discography using indigo carmine, an angled working sheath (diameter = 7 mm) and endoscope were inserted in the superior articular process (SAP), and the muscle attached to the lateral margin of the SAP was carefully detached and electrocoagulated. The exposed SAP was removed using a 3.5 mm diameter high-speed drill (NSK-Nakanishi Japan, Tokyo, Japan) until the markedly stained disc space was adequately exposed (medial lateral distance > 13 mm). We removed the disc material mainly using forceps. After the corresponding disc space was cleaned up, the angled working sheath and endoscope were replaced with dilators especially designed for cage insertion (Figure 2a). Using these dilators, the appropriate height for the cage was determined. At this step, the skin incision was enlarged to 15 mm, and the disc height was increased with the expansion of the PPS. After the insertion of the outer sheath (Figure 2b) and demineralized bone matrix grafting (Grafton^TM^, Medtronic Sofamor Danek, Memphis, TN, USA), the outer sheath was replaced with sliders originally designed for cage insertion (Figure 2c,d). The 12 mm-wide cage (ReyKamCage^®^ Robert Reid Inc., Tokyo, Japan) was carefully inserted under fluoroscopic guidance (Appendix A). The sliders were removed, and the wound was closed using 1–2 hypodermic sutures without a drainage tube. Finally, the PPS was placed on the contralateral side, and the rods were compressed and fixed with appropriate force on both sides.

## 3. Results

The detailed characteristics of each case are summarized in Table 1. This technical report consisted of 12 patients (male: 7, female: 5). The major symptom was radiculopathy, noted in eight patients; cauda equina syndrome was noted in three patients, and lower back pain in one patient. The mean age at surgery was 68.3 years (range, 47–84 years). The affected vertebral level was L3/4 in three patients, L4/5 in five patients, and L5/S1 in four patients.

The mean operation time was 109.4 min (range, 73–160 min), and the mean duration of postoperative hospital stay was 7.7 days (range, 5–15 days). The leg pain improved significantly in 10/12 cases (mean NRS score changed from 6.5 to 1.2, *p* < 0.01). The major symptom in Case No. 6, where a change in the NRS score was not observed, was cauda equina syndrome (walking disturbance and paresthesia of both legs), and the patient was aware of an improvement in dysuria after FELIF. The major symptom in Case No. 12, in which an increase in the NRS score was noted, was lower back pain, and the patient complained of pain around the left knee 4 days after the FELIF. The pain improved following a left L4 nerve root block, and the patient has been conservatively followed up at our outpatient clinic. Although these two patients still complain of moderate pain, intraoperative complications, such as exiting nerve injury or dural tears, were not observed.

The mean follow-up period was 6.2 months (range, 2–10 months). Although slight cage subsidence was observed in Case No. 1 and 2, in which a 10 mm height cage was used during this period, no symptom related to subsidence, such as radiculopathy, was observed.

With respect to the Brandner’s disc index, both the anterior and posterior disc indices (B/A and C/A, as shown in Figure 1) increased significantly, from 0.18 ± 0.12 to 0.32 ± 0.13 (*p* < 0.01) and from 0.08 ± 0.05 to 0.23 ± 0.05 (*p* < 0.01), respectively (Figure 3). The increase in the disc height was confirmed in the plain radiographs and CT scans in all cases. A typical change observed in a CT scan is shown in Figure 4 (Case No. 5).

## 4. Discussion

Uniportal FESS was originally developed for the treatment of lumbar disc herniation and has recently been used for LIF [6,7,8,9,10,11,12,13,14,15,16,17,18,19]. Owing to technical refinements and the development of new instruments, FESS can be performed for a wide spectrum of diseases. However, only a few studies have reported on LIF using uniportal FESS for the treatment of LDD. Furthermore, studies analyzing disc height changes have not been reported. We therefore retrospectively analyzed the changes in disc height following the newly established LIF procedures.

Our analysis indicates that the postoperative disc height was significantly greater after FELIF. Although the mean NRS score changed from 6.5 to 1.2, when focusing only on the eight cases of foraminal LSCS, the mean NRS changed from 7.1 to 0.1 (except for one case, where the leg pain disappeared immediately after the operation). Furthermore, a satisfactory response was obtained in the initial two cases (Case No. 2 and 3) with central LSCS with cauda equina syndrome, but not obtained in Case No. 6. We could achieve indirect decompression of the central LSCS (Figure 5) but could not achieve patient satisfaction in this case. As our experiences of FELIF were very few, we have not yet found factors to distinguish FELIF-responder from cauda equine syndrome. Therefore, we should limit the indication for FELIF to LDD with disc height loss leading to corresponding radiculopathy at this moment. In contrast, 10 out of 12 patients had a history of previous spinal surgery (Table 1). In addition to the minimal invasiveness of FELIF, FELIF may serve as a potential alternative strategy for overcoming the failure of previous spinal surgeries. This indicates that FELIF may contribute towards reducing the rate of failed back surgery syndrome, which has recently been the most concerning pathophysiological state since the advent of spinal surgery.

With respect to the operation time, the mean operation time for our initial 12 cases was 109.4 min. As the operation time of endoscope-assisted LIF, a standard approach in our hospital, was 102 (range, 59–162) min [24], 109.4 min was not significantly different from the operative time for the conventional approach. Likewise, the mean operation time was not inferior to that reported in previous FELIF studies (84.5–285.7 min) [6,7,8,9,10,11,12,13,14,15,16,17,18,19]. Presently, we only perform FELIF for patients with LDD with disc height loss; the removal of the disc material is not a time-consuming step. If we perform FELIF for patients with other spinal diseases without disc height loss, this step will be one of the factors prolonging the operation time. The further development of endoscopic instruments for rapidly removing the disc material may aid in preventing the prolongation of the operative time.

With respect to complications, it is possible to reduce the incidence of dural tears and postoperative hematoma in patients undergoing FELIF. As it is not necessary to expose the dural sac and there is a clear operative field under saline irrigation in FELIF, surgeons can easily confirm the active bleeding site. The most critical complication of FELIF is exiting nerve root injury, and newly developed sliders for cage-insertion can protect the exiting nerve root. The enlargement of Kambin’s safety triangle using a high-speed drill is another key step for avoiding this injury [25]. If we acquire enough disc space (medial lateral width > 13 mm), we can safely insert the cage. Preoperative radiological examination is also important for preventing complications. If the presence of a conjoined nerve root or large artery around the vertebral foramen (e.g., variation of segmental artery) is suspected [26], we should avoid a transforaminal approach and select other alternative operative approaches.

Recently, Nagahama et al. reported excellent clinical outcomes following transforaminal LIF using a 3.9 mm working channel endoscope (SpineTIP^®^ Transforaminal Approach Kit, Karl Storz GmbH, Tuttlingen, Germany) [17]. They treated 25 cases of degenerative spondylolisthesis of the L4 vertebra; the spondylolisthesis was corrected using a PPS system before endoscopic manipulation. They did not apply their procedure to the spondylolisthesis of the L5 vertebra, although it is theoretically possible, despite the vertebral level affected. The adequate removal of the SAP at the L5/S1 level may be the important operative step for avoiding exiting nerve injury. As their procedure was very similar to our procedure and suitable for Japanese circumstances, these procedures are likely to be widely used by Japanese spine surgeons. Elaborating the operative steps other than the endoscopic manipulation, such as the manipulation of the PPS system shown by Nagahama, may expand the operative indications for FELIF to other degenerative lumbar diseases presenting with spondylolisthesis or instability.

We could not indicate the final fusion status in this technical report because of the short follow-up periods. We had not confirmed both distinct intervertebral fusion (Grade 1 of Bridwell fusion grading system) and pseudoarthrosis. However, the tomographic roentgenography revealed that Grade 2 of the Bridwell fusion grading system (not fully remodeled and incorporated, but no lucency present) was achieved at least 6 months after FELIF (Figure 6). We therefore consider that these cases are under the process of intervertebral fusion.

### Study Limitations

This study has several limitations. First, the sample size was too small to conclude the definitive efficacy of this new procedure. Second, the follow-up duration was not sufficient to evaluate the long-term effect and safety. Third, tomographic roentgenography or CT at the appropriate time would also be required for the definitive confirmation of vertebral fusion. These radiographical inspections are required for all cases at least 6–12 months after FELIF.

## 5. Conclusions

The preliminary results over a short follow-up period showed that the operation time and the intraoperative safety of 4.1 mm working channel FELIF were not inferior to those of previously reported FELIF. Radiculopathy caused by foraminal stenosis was the most effective target for FELIF. The adequate removal of the superior articular process is an important operative step for avoiding exiting nerve injury, especially for FELIF at the L5/S1 level.

## Figures and Tables

**Figure 1 medicina-56-00478-f001:**
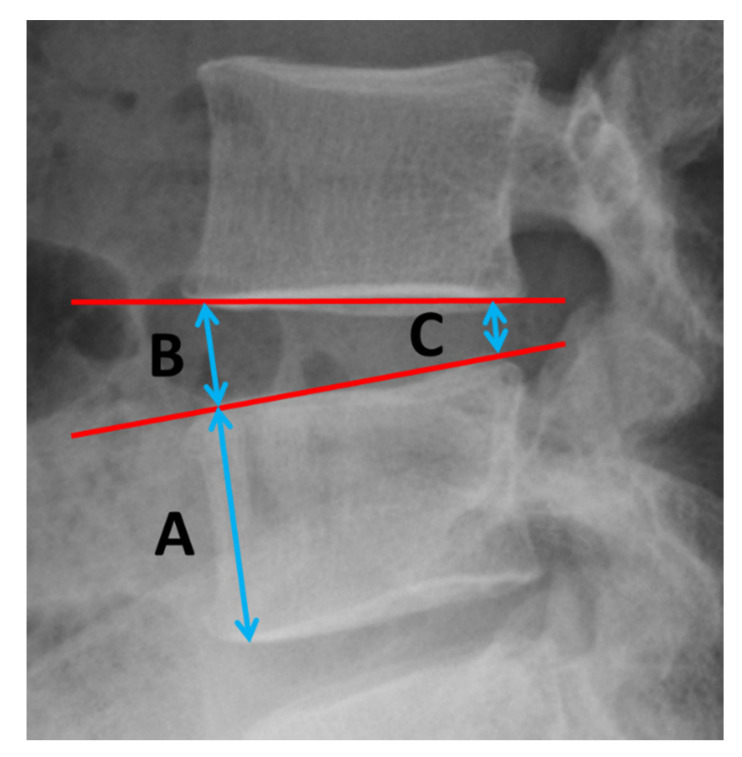
Brandner’s disc index. A is the maximum height of the adjacent vertebral body. B and C are the disc heights at its anterior and posterior positions. B/A = anterior disc index; C/A = posterior disc index.

**Figure 2 medicina-56-00478-f002:**
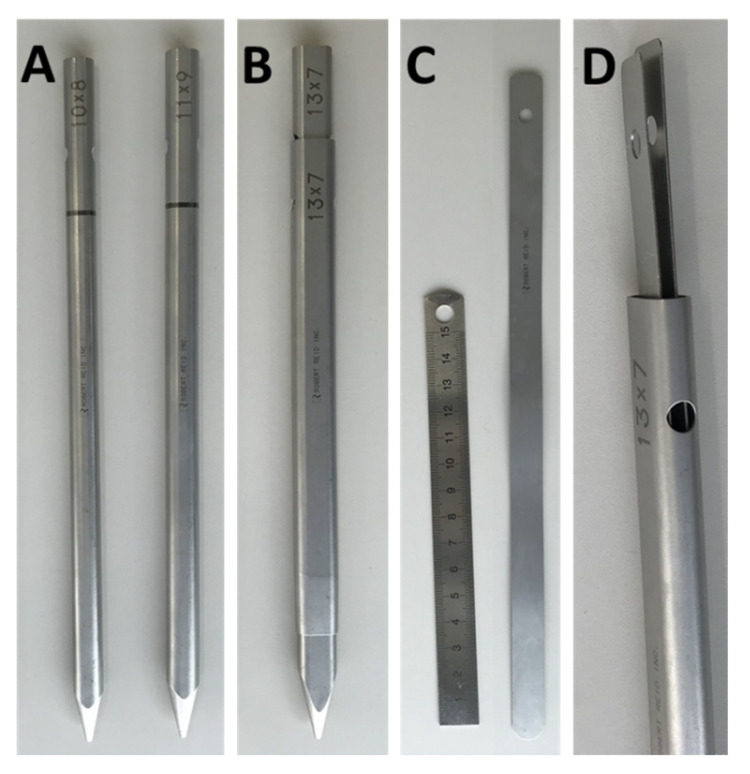
Specially designed instruments for cage insertion in FELIF. (**A**) Serial dilators (8–11 mm) for the determination of the appropriate cage size. Each dilator can fit two different disc heights with a 90° rotation. (**B**) Outer sheath and the exclusive dilator. (**C**) Slider for cage insertion. (**D**) Two sliders are used to locate the disc space through the outer sheath. FELIF = full-endoscopic lumbar interbody fusion.

**Figure 3 medicina-56-00478-f003:**
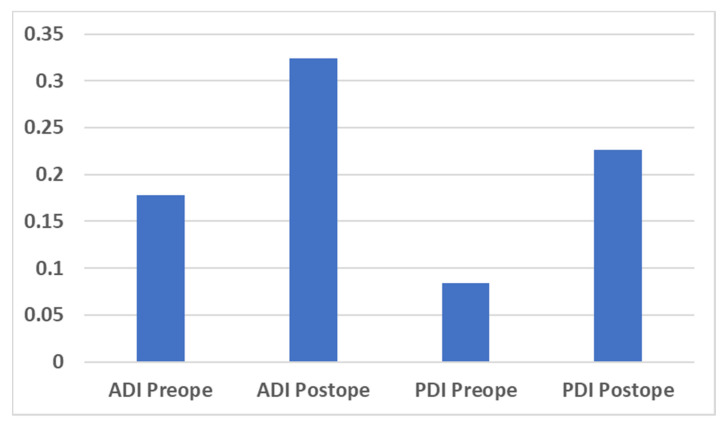
Bar chart for the preoperative and postoperative Brandner’s disc index. ADI = anterior disc index; PDI = posterior disc index; Preope = preoperative; Postope = postoperative.

**Figure 4 medicina-56-00478-f004:**
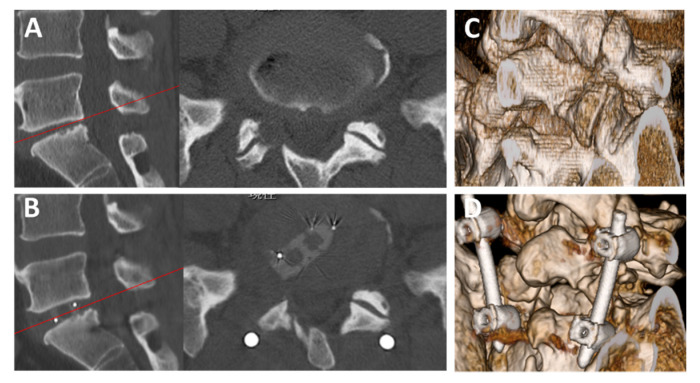
Preoperative and postoperative CT images. (**A**) Preoperative sagittal (left) and axial (right) CT findings of L5/S1 FELIF (Case No. 5, a 47-year-old man). (**B**) Postoperative sagittal (left) and axial (right) CT findings. The red lines in the sagittal images indicate the level of axial scanning. Note that the disc height increased after FELIF. (**C**,**D**) Preoperative (**C**) and postoperative (**D**) 3-dimensional CT findings. Note that an appropriate extent of the SAP was removed for cage insertion. CT = computed tomography; FELIF = full-endoscopic lumbar interbody fusion; SAP = superior articular process.

**Figure 5 medicina-56-00478-f005:**
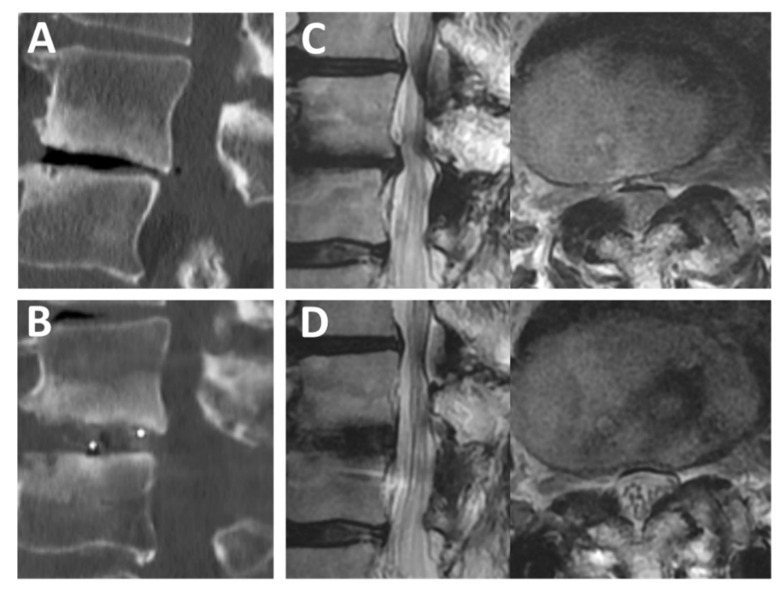
Preoperative and postoperative CT and MRI scans. (**A**,**B**) Preoperative (**A**) and postoperative (**B**) sagittal CT findings of L3/4 FELIF (Case No. 6, a 74-year-old man). (**C**) Preoperative sagittal (left) and axial (right) MRI findings. (**D**) Postoperative sagittal (left) and axial (right) MRI findings. Note that cauda equina can be observed separately after FELIF (**D**, right). CT = computed tomography; FELIF = full-endoscopic lumbar interbody fusion; MRI = magnetic resonance imaging.

**Figure 6 medicina-56-00478-f006:**
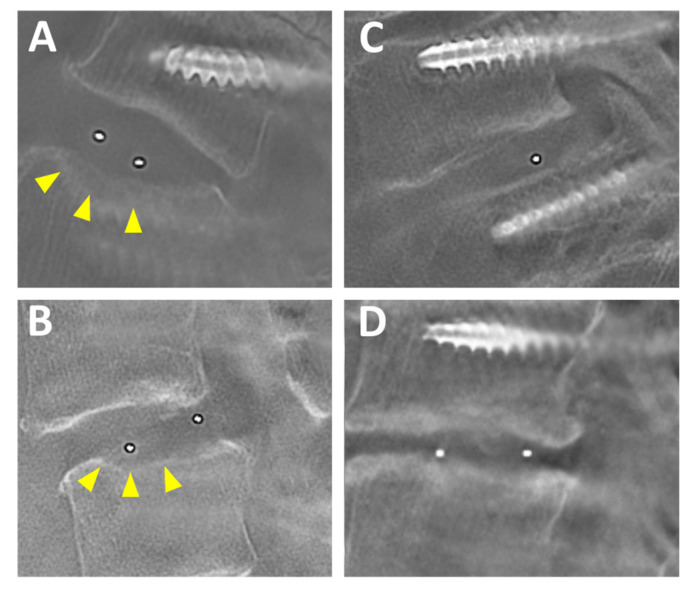
Preoperative tomographic roentgenography. (**A**) Case No. 1, an 83-year-old woman. (**B**) Case No. 2, a 64-year-old woman. (**C**) Case No. 3, an 84-year-old woman. (**D**) Case No. 3, a 75-year-old woman. Note that no lucency is present at the top and bottom of the cage. Slight cage subsidence was observed in Case No. 1 and 2 (arrow heads).

**Table 1 medicina-56-00478-t001:** Summary of the 12 cases.

Major Symptom *	Level	Approach *	Cage Size	Operative Time (min)	Hospital Stay (Days)	Follow-Up Periods (m)	NRS Preope	NRS Postope	Previous Operation **	Interval (m)
L L4 radiculopathy	L4/5	L	10 × 12 × 30 mm	106	9	10	6	0	L4/5 FEL	4
cauda equina syndrome	L4/5	R	10 × 12 × 35 mm	88	7	9	7	1		
cauda equina syndrome	L4/5	L	8 × 12 × 35 mm	91	7	8	8	2	L4/5 MEL	13
R L4 radiculopathy	L4/5	R	8 × 12 × 35 mm	114	8	8	6	1	L4/5 FED	6
R L5 radiculopathy	L5/S1	R	8 × 12 × 30 mm	151	8	7	8	0	L5/S1 FEL	9
cauda equina syndrome	L3/4	R	8 × 12 × 35 mm	80	7	6	5	5	L4/5-L5/S1 PLF	18
L L5 radiculopathy	L5/S1	L	8 × 12 × 35 mm	82	7	6	8	0	L5/S1 Open discectomy	13
R L4 radiculopathy	L4/5	R	8 × 12 × 35 mm	140	5	5	10	0	L3/4/5 MEL	62
L L4 radiculopathy	L3/4	L	8 × 12 × 35 mm	73	7	5	7	0	L3/4 MEL	10
R L5 radiculopathy	L5/S1	R	8 × 12 × 30 mm	131	6	2	4	0	L5/S1 FEL	13
L L5 radiculopathy	L5/S1	L	8 × 12 × 35 mm	97	6	2	8	0	L5/S1 FEL	8
low back pain	L3/4	L	8 × 12 × 35 mm	160	15	6	1	5		

* R/L = right/left; ** FEL = full-endoscopic laminectomy; MEL = microendoscopic laminectomy; FED = full-endoscopic discectomy; PLF = posterior lumbar fusion.

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
