# Peer review of "Full-Endoscopic Lumbar Interbody Fusion for Treating Lumbar Disc Degeneration Involving Disc Height Loss: Technical Report"

_medicina, 2020, doi:10.3390/medicina56090478_

Round 1

Reviewer 1 Report

The authors reported a case series of 12 patients underwent full-endoscopic lumbar interbody fusion with a 4.1-mm working channel endoscope.

The authors should clarify the following:

1) How long was the follow-up?

2) Any cage subsidence during follow up?

3) Add a paragraph in the discussion regarding fusion rate of full endoscopic fusion vs MIS tubular approach.

Author Response

Iwai FESS Clinic 

8-18-4 Minamikoiwa Edogawa-ku Tokyo, 133-0056 Japan

August 28, 2020

Ms. Ref. No.:  medicina-903246

Dear Editor-in-Chief, Medicina.

We thank you and the reviewers for the insightful critique of our manuscript entitled “Full-endoscopic lumbar interbody fusion for treating lumbar disc degeneration involving disc height loss: initial clinical experience.” We appreciate the fact that you and the reviewers have given us the opportunity to submit a revised manuscript and that it will be considered for publication in Medicina.

The manuscript was carefully checked and rewritten according to editor’s and the reviewers’ comments and suggestions. We would like to address those comments point by point, as follows:

  • Reviewer #1 suggested to indicate follow-up periods. According to this suggestion, we indicated each follow-up period in Table 1 and described that in results section.
  • Reviewer #1 also suggested to indicate cage subsidence during follow-up period. We only observed slight cage subsidence in Case 1 and 2. According to this suggestion, we described this in results section.
  • Reviewer #1 requested to add a paragraph in the discussion regarding fusion rate of full-endoscopic fusion vs MIS tubular approach. According to this suggestion, we added new paragraph.
  • Reviewer #2 suggested that this paper is more suitable for categories for case report or surgical techniques than original article. According to this suggestion, we changed the category for “Technical report”.
  • Reviewer #2 expressed concern about inappropriate usage of MISS techniques for cauda equine syndrome. This concern is quite reasonable. We applied FELIF to second and third cases with cauda equine syndrome (Case No. 2 and 3), and obtained good clinical outcome. We therefore applied FELIF to another case (Case No. 6) with cauda equine syndrome. We obtained indirect decompression radiologically but not obtained clinical improvement. As these were initial experiences of our FELIF, we have not yet found factors to distinguish FELIF-responder from cauda equine syndrome. To solve reviewer’s concern, we added the sentences in Discussion section.
  • Reviewer #2 also expressed concern that the number of patients and the patient follow-up period are too short to determine a definite clinical outcome. This concern is also quite reasonable. The policy of our hospital toward spinal diseases is to treat without instrumentation as much as possible. Especially for foraminal stenosis, full-endoscopic spine surgery is the first choice (Journal spine surgery 2018, 4(3):594-601). Therefore, to accumulate large number of patients received lumbar interbody fusion is difficult. Please understand this situation. We added exact follow-up periods and rewrote the article as “Technical report”.
  • Reviewer #2 suggested to define fusion status. The longest follow-up period is 10 months and we have not yet confirm both distinct intervertebral fusion (Grade 1 of Bridwell fusion grading system) and pseudoarthrosis. We consider these cases are under the process of intervertebral fusion. According to this suggestion, we revised discussion section and added Figure 6 for the postoperative tomographic roentgenography.
  • Similar to Reviewer #2, reviewer #3 pointed out that a report of 12 patients is very few. According to this suggestion, we rewrote the article as “Technical report”.
  • Similar to Reviewer #2, reviewer #3 suggested that 6-month and 1-year clinical and radiography outcomes should be reported for a case-series regard fusion patients. CT scans and tomogram in X-ray at 6-month after FELIF have already finished on 8 out of 12 cases but the distinct intervertebral fusion (Grade 1 of Bridwell fusion grading system) was not observed. The follow-up period is too short to determine fusion rate for FELIF. According to this suggestion, we revised discussion section.
  • Along with above revisions, we added a few references for better understanding of readers.
  • In addition to above points, we found mistake in Table 1 regarding age of Case No. 12. We correct this mistake and related text (change 60 to 63).

A great deal of effort has gone into revising the manuscript, and we concur with the editor and reviewers that these revisions have significantly strengthened the manuscript. We deeply appreciate the editor’s invitation to submit the revised manuscript. We hope this revised version is satisfactory to the reviewers and is suitable for publication in Medicina.

Please send all correspondence to:

Hisashi KOGA, MD, PhD

Department of Neurosurgery, Iwai FESS Clinic 

8-18-4 Minamikoiwa Edogawa-ku Tokyo, 133-0056 Japan

TEL: +81-3-5694-4976 

FAX: +81-3-5694-1010

We eagerly look forward to your favorable response.

Sincerely yours,

Hisashi KOGA, M.D., Ph.D.

Department of Neurosurgery, Iwai FESS Clinic

Reviewer 2 Report

The submitted article “Full-endoscopic lumbar interbody fusion for treating lumbar disc degeneration involving disc height loss: initial clinical experience” demonstrated the clinical experiences of 12 consecutive patients and suggested clinical usefulness of FELIF for treating LDD.

At the present time when endoscopic assisted spine surgeries are becoming popular, the submitted article appear to be meaningful in sharing the clinical results of the suggested surgical technique. However, it is questionable whether the results can give the conclusion in abstract that the authors insist, because of fragmentary clinical experiences in retrospective manner.

There are my concerns as bellows

  1. This submitted paper is more suitable for categories for case report or surgical techniques than original article.
  2. It seems that the authors want to emphasize the indirect decompression effects through the Kambin’s triangle by FELIF. However, they didn’t demonstrate the results of control data. Also, the preoperative baseline characteristics for included patients were not homogeneous. Such kinds of MISS techniques are usually not appropriate for cauda equine syndrome, but they included 3 patients with CES.
  3. The number of patients and the patient follow-up period are too short to determine a definite clinical outcome. Maybe, the latest patient was enrolled 1 month ago.
  4. Also, they didn’t define fusion status. It is doubt that the operated segment can be fused from the figures presented

Author Response

(The authors gave the same response as above.)

Reviewer 3 Report

The authors present a case series of 12 patients who underwent fully endoscopic lumbar interbody fusion using a transforaminal approach to the disc space. 

The major symptoms of the patient included in this study were low back pain, unilateral radiculopathy, and cauda equina.

This is an emerging technology and surgical technique that is very exciting for the field. However, a report of 12 patients is very few for a publication in the spine literature. Additionally, 6-month and 1-year clinical and radiography outcomes should be reported for a case-series regard fusion patients.

Author Response

(The authors gave the same response as above.)

Round 2

Reviewer 2 Report

There are no further arguments on the aemndments I pointed out

Overall, there is no disagreement with publication processing in the form of a technical reports

Author Response

Ms. Ref. No.:  medicina-903246

Dear Editor-in-Chief, Medicina.

Thank you so much for sparing reviewer’s precious time for second round reviewing our article.

Reviewer #3 suggested that following conclusion was not proven: “The preliminary results over a short follow-up period showed that the operative outcomes of 4.1-mm working channel FELIF were not inferior to those of conventional LIF for the treatment of LDD.” This is quite appropriate suggestion.

According to this suggestion, we changed above sentence as follows: The preliminary results over a short follow-up period showed that the operation time and the intraoperative safety of 4.1-mm working channel FELIF were not inferior to those of previously reported FELIF.

We deeply appreciate the reviewer’s insightful suggestions. We hope this revised version is satisfactory to the reviewers and is suitable for publication in Medicina.

Hisashi KOGA, MD, PhD

Reviewer 3 Report

I appreciate the authors concerns and difficulty obtaining more patients than the 12 presented. However, the final conclusion of the paper is “The preliminary results over a short follow-up period showed that the operative outcomes of 4.1-mm working channel FELIF were not inferior to those of conventional LIF for the treatment of LDD.” This reviewer does not believe that this conclusion was proven by what was presented in this manuscript, as there was no true comparison to a conventional LIF group; just a literature discussion.

Author Response

(The authors gave the same response as above.)
